# The Formulation and Characterization of Wound Dressing Releasing S-Nitrosoglutathione from Polyvinyl Alcohol/Borax Reinforced Carboxymethyl Chitosan Self-Healing Hydrogel

**DOI:** 10.3390/pharmaceutics16030344

**Published:** 2024-02-29

**Authors:** Juliana Palungan, Widya Luthfiyah, Apon Zaenal Mustopa, Maritsa Nurfatwa, Latifah Rahman, Risfah Yulianty, Nasrul Wathoni, Jin-Wook Yoo, Nurhasni Hasan

**Affiliations:** 1Faculty of Pharmacy, Hasanuddin University, Jl. Perintis Kemerdekaan KM 10, Makassar 90245, Indonesia; palunganjuliana@gmail.com (J.P.); luthfiyahwidya@gmail.com (W.L.); tifah_rahman15@yahoo.com (L.R.); risfahyulianty@unhas.ac.id (R.Y.); 2Research Center for Genetic Engineering, National Research and Innovation Agency (BRIN), Bogor 16911, Indonesia; azae001@brin.go.id (A.Z.M.); maritsa.nurfatwa@gmail.com (M.N.); 3Faculty of Pharmacy, Universitas Padjajaran, Jl. Ir. Soekarno KM 21, Jatinagor 45363, Indonesia; nasrul@unpad.ac.id; 4College of Pharmacy, Pusan National University, Busan 46241, Republic of Korea; jinwook@pusan.ac.kr

**Keywords:** wound dressings, self-healing hydrogel, S-Nitrosoglutathione, polyvinyl alcohol, borax, carboxymethyl chitosan, physicochemical properties

## Abstract

Self-healing hydrogels often lack mechanical properties, limiting their wound-dressing applications. This study introduced S-Nitrosoglutathione (GSNO) to self-healing hydrogel-based wound dressings. Self-healing hydrogel mechanical properties were improved via polymer blends. Applying this hydrogel to the wound site allows it to self-heal and reattach after mechanical damage. This work evaluated polyvinyl alcohol (PVA)-based self-healing hydrogels with borax as a crosslinking agent and carboxymethyl chitosan as a mechanical property enhancer. Three formulations (F1, F4, and F7) developed self-healing hydrogels. These formulations had borax concentrations of 0.8%, 1.2%, and 1.6%. An FTIR study shows that borate ester crosslinking and hydrogen bonding between polymers generate a self-healing hydrogel. F4 has a highly uniform and regular pore structure, as shown by the scanning electron microscope image. F1 exhibited faster self-healing, taking 13.95 ± 1.45 min compared to other formulations. All preparations had pH values close to neutrality, making them suitable wound dressings. Formula F7 has a high drug content (97.34 ± 1.21%). Good mechanical qualities included high tensile stress–strain intensity and Young’s modulus. After 28 h of storage at −20 °C, 5 °C, and 25 °C, the self-healing hydrogel’s drug content dropped significantly. The Korsmeyer–Peppas release model showed that the release profile of GSNO followed Fickian diffusion. Thus, varying the concentration of crosslinking agent and adding a polymer affects self-healing hydrogels’ physicochemical properties.

## 1. Introduction

Nitric oxide (NO) is a highly reactive chemical and a crucial messenger in the body’s signaling processes. NO has various important physiological roles, such as neurotransmission, wound healing, regulation of blood pressure, platelet adhesion, immunological response, and antibacterial and antibiofilm activities [1]. In addition to their antibacterial and antibiofilm properties, these chemicals can be utilized for wound healing by controlling inflammation, widening blood vessels, promoting the formation of new blood vessels, stimulating cell growth, and facilitating the deposition of extracellular matrix [2,3]. Nevertheless, the therapeutic application of NO is limited by its gaseous form and short half-life of 3–4 seconds. Therefore, the use of exogenous NO donors is necessary to address these characteristics of NO [1]. Therefore, S-Nitrosoglutathione (GSNO) is employed as an NO donor. GSNO is commonly employed as an exogenous NO donor due to its biocompatible properties and ease of purification through precipitation and drying. This makes it more stable compared to other NO donors [4]. GSNO will undergo conversion into glutathione (GSH), spontaneously release NO, and contribute to the healing of the wound [5]. The incorporation of GSNO into wound dressing is highly anticipated to facilitate effective wound healing.

Studies have been conducted on the advancement of GSNO delivery systems for use as wound dressings, including hydrogels, ointments, films, and micro/nanoparticles. Hydrogel is commonly employed as a wound dressing due to its ability to maintain moisture in the wound area, creating ideal circumstances for effective wound healing [1]. Furthermore, GSNO exhibits high solubility in water, making it well-suited for topical and targeted administration in hydrophilic matrices [6]. Nevertheless, hydrogels have inadequate mechanical characteristics and experience deformation when subjected to external pressures. A self-healing hydrogel has been created, capable of automatically repairing itself upon encountering an external force [7]. Self-healing hydrogel is an appealing biomaterial that also possesses the additional benefit of being capable of conforming to the shape, size, and imperfections of the area where it is applied [8].

Thus, research will be conducted on the development of GSNO with a self-healing hydrogel using a Taguchi orthogonal array design-optimized polyvinyl alcohol (PVA)-borax polymer reinforced with carboxymethyl chitosan (CmChi). The Taguchi method is an experimental design that enables the selection or regulation of more consistent and optimal products or processes in order to reduce variation by minimizing the influence of noise and effects. PVA is a polymer that is extensively employed as a hydrogel matrix due to its non-toxic nature, high hydrophilic properties, biodegradability, and biocompatibility [9]. Consequently, it finds utility in biomedical applications. According to research by Zhang et al., a self-healing hydrogel that creates crosslinks can be produced by utilizing a single PVA that is frozen and thawed. A hydrogel preparation with low transparency, however, necessitates a high PVA content, substantial energy consumption, and an extended freeze–thaw cycle [10]. Therefore, the utilization of reversible binders, such as borax, in the preparation of self-healing PVA hydrogels results in the development of hydrogels that exhibit notably enhanced flexibility and ductility. However, it has poor stability, limited service life, and feeble mechanical properties [11].

Hydrogels synthesized from natural polysaccharide polymers, like chitosan, hyaluronic acid, sodium alginate, and dextran, are excellent substances for drug delivery due to their favorable biocompatibility and biodegradability properties. The study conducted by Lin et al. indicates that including chitosan in the PVA/dextran hydrogel enhances its mechanical properties, resulting in improved strength and elongation [12]. Consequently, the dressing remains undamaged, even during limb movement when the hydrogel is applied [13]. Nevertheless, the primary limitation of chitosan lies in its inadequate water solubility. Carboxymethyl chitosan (CmChi) is a frequently encountered derivative of chitosan. Due to its favorable biocompatibility, biodegradability, hydrophilicity, antimicrobial characteristics, functional amino groups, and water solubility, this substance is extensively employed in the biomedical domain [12,14]. Polyvinyl alcohol (PVA) is commonly blended with other polymers, such as CmChi, due to its excellent compatibility with PVA in water-based solutions. The hydroxyl group on PVA and the hydroxyl group, amino group, and carboxyl group on CmChi can create intermolecular hydrogen bonds, resulting in a substantial enhancement in the mechanical characteristics of the produced preparations [15,16]. In their study published in 2023, Zhang et al. found that including CmChi in the PVA matrix enhances its biocompatibility and contributes to its favorable mechanical characteristics and water adsorption at room temperature [17]. Based on these findings, we hypothesized that the addition of polymer (CmChi) in the PVA–borax crosslinking could increase the mechanical properties of the resulting self-healing hydrogels.

In the present work, we aimed to develop NO-releasing self-healing hydrogels as novel hydrogel-based wound dressings. This work evaluated PVA-based self-healing hydrogels with borax as a crosslinking agent and CmChi as a mechanical property enhancer. PVA and CmChi will be crosslinked with the addition of borax as a crosslinking agent. After optimizing the formulation, their physicochemical properties were evaluated, including morphological analyses by SEM, self-healing behavior, mechanical properties, pH, water uptake, and drug content. Finally, stability tests and in vitro drug release were examined to determine the self-healing hydrogel’s ideal properties as a wound dressing.

## 2. Materials and Methods

### 2.1. Materials

Reduced L-glutathione (MW = 307.32, 98%) was purchased from Shanghai Aladdin Biochemichal Technology (Chuhua Branch Road, Shanghai). Polyvinyl alcohol (PVA, MW = 44.05, 87–89%), *N*-carboxymethylchitosan (CmChi, MW ~400,000 Da, >95%), sodium nitrite (NaNO_2_), methylene blue hydrate, potassium bromide (KBr), borax, and phosphate-buffered saline (Dulbecco A) Oxoid BR0014G were purchased from Sigma-Aldrich (St. Louis, MO, USA). Cellophane visking tubing dialysis osmosis membrane was obtained from Medicell Membrane Ltd. (London, UK). All the other reagents and solvents were of the highest analytical grade.

### 2.2. Synthesis of GSNO

The synthesis of GSNO was conducted using the methodology described by Hasan et al., with minor adjustments. To prepare the solution, 1.53 g of reduced L-GSH was dissolved in 2.5 mL of 2 M hydrochloric acid at a temperature of 4 °C. This resulted in a solution with a final concentration of 0.625 M. Subsequently, a solution containing 0.345 g of NaNO_2_, which had been previously dissolved in 5.5 mL of distilled water, was introduced. The resulting mixture was then subjected to stirring for a duration of 50 min using a magnetic stirrer, while being kept in an ice bath. Subsequently, cold diethyl ether was introduced and agitated for a duration of 50 min using a magnetic stirrer. The resulting solid was subsequently isolated through filtration with an oil-less vacuum pump (Rocker 300, Rocker, Taipei, Taiwan) and subjected to three rounds of rinsing with cold diethyl ether. The pink GSNO was dried out for 24 h and thereafter stored at a temperature of −20 °C for subsequent studies [18].

### 2.3. Preparation of Self-Healing Hydrogels

The self-healing hydrogel was synthesized by chemically bonding PVA and borax with CmChi in an aqueous solution. The Taguchi Orthogonal array design was used to optimize the formula of the self-healing hydrogel. All the concentrations of polymers and borax followed the concentrations in Table 1. Initially, for the F1, the PVA (4% wt) was dissolved in water by vigorously stirring it continuously at a temperature of 90 °C in a water bath until the PVA was fully dissolved. Borax (0.8% wt) was dissolved in water at a temperature of 90 °C using a water bath until complete dissolution occurred. Subsequently, CmChi (1.25% wt) and GSNO (1% wt) were combined and solubilized in water, then introduced into the PVA solution and subjected to homogenization (Homogenizer, WiseStir^®^ HS-50A, Daihan, Korea). Gradually, the borax solution was incorporated into the mixture while thoroughly blending it to create the self-healing hydrogel known as PVA-B-CmChi/GSNO (Figure 1). The entire process of synthesizing the self-healing hydrogel preparation was conducted under light-protected conditions. The blank self-healing hydrogel without GSNO (PVA-B-CmChi) was prepared similarly without the addition of GSNO.

### 2.4. Physicochemical Characterization of the Self-Healing Hydrogel

#### 2.4.1. Organoleptic 

The blank self-healing hydrogel (PVA-B-CmChi) and the NO-releasing self-healing hydrogel (PVA-B-CmChi/GSNO) were visually observed, including shape, size, and color.

#### 2.4.2. Scanning Electron Microscopy (SEM)

The morphology of PVA-B-CmChi and PVA-B-CmChi/GSNO self-healing hydrogels were analyzed using SEM (Jeol^®^ IT 200, Tokyo, Japan). The self-healing hydrogels were mounted on carbon tape before being vacuum-coated with platinum for 2 min. The sample morphology was then observed at accelerating voltages ranging from 2 to 5 KV [19].

#### 2.4.3. Fourier Transform Infrared Spectroscopy (FTIR) Characterization

The spectra of PVA-B-CmChi and PVA-B-CmChi/GSNO self-healing hydrogels were observed using FTIR (Shimadzu^®^ IRPrestige-21, Tokyo, Japan). A total of two mg of sample were mixed with 200 mg of KBr. Then, this was grounded and pressed into a disc for analysis. The spectrum was then recorded at wavenumbers 4000 to 400 cm^−1^ with a resolution of 4 cm^−1^ [20].

#### 2.4.4. Self-Healing Behavior of PVA-B-CmChi and PVA-B-CmChi/GSNO

ThePVA-B-CmChi and PVA-B-CmChi/GSNO self-healing hydrogels were cut into two parts; one part was stained with methylene blue, and the other part was not dyed to differentiate the two pieces. The two ends of the self-healing hydrogels were then brought close together, and the time needed for the self-healing hydrogel to become one unit was observed and counted [20].

#### 2.4.5. Mechanical Characterization of Self-Healing Hydrogel

The tensile test was measured using a microcomputer-controlled electronic universal testing machine (Zhiqu^®^ digital display tension meters, Dongguan, China) at a crossheaving speed of 100 mm min^-1^ at a temperature of 25 °C. The self-healing hydrogel PVA-B-CmChi and PVA-B-CmChi/GSNO were cut into cubes with dimensions of 4 mm × 3 mm × 3 mm and mounted on a tool, and the force used to pull the self-healing hydrogel preparation was measured. The stress (σ) is calculated using the equation F/A, where F is the load force, and A is the surface area. Strain (ε) shows the deformation of the material under a predetermined external force, calculated by the equation l/l0 × 100%, where l0 and l are the lengths of the sample before and after deformation. Young’s modulus (E) is calculated by the slope of the stress–strain curve in the linear range [20].

#### 2.4.6. pH of Self-Healing Hydrogel

The pH of the PVA-B-CmChi and PVA-B-CmChi/GSNO self-healing hydrogels was measured using a pH meter (PL-700 Series Bench Top pH Meter, Taipei, Taiwan). A total of one gram of self-healing hydrogel was dissolved in distilled water using a magnetic stirrer until the self-healing hydrogel was completely dissolved. Then, the pH of the self-healing hydrogel was measured using a pH meter [21].

#### 2.4.7. Loading Efficiency (LE) and Loading Capacity (LC)

The PVA-B-CmChi/GSNO were measured for LE and LC. In addition, to determine whether there was decomposition of GSNO in the self-healing hydrogel during storage, the LE and LC of the self-healing hydrogels were measured. The PVA-B-CmChi/GSNO self-healing hydrogel was weighed at 40 mg and placed in cold water to prevent further GSNO decomposition. The sample was then crushed and dispersed in solution using a magnetic stirrer. After centrifugation for 20 min, the clear supernatant was analyzed spectrophotometrically at a wavelength of 335 nm to determine the GSNO content in the self-healing hydrogel. The LE and LC values are calculated based on the formula [22]:$$\%LE=\frac{Amount\;of\;GSNO\;in\;hydrogel}{Initial\;amount\;of\;drug\;loaded}\times 100\;\%LC=\frac{Amount\;of\;GSNO\;in\;hydrogel}{Total\;amount\;of\;polymer\;and\;GSNO}\times 100$$

#### 2.4.8. Swelling Ratio of Self-Healing Hydrogel

Several freeze-dried PVA-B-CmChi and PVA-B-CmChi/GSNO self-healing hydrogels were immersed in simulated wound fluid (SWF) at a temperature of 25 °C and moved for a predetermined period. Surface water was then taken with the help of filter paper, and the swollen self-healing hydrogel was then weighed, and the percentage of water absorption capacity was measured using the equation: $$S\;\left(\%\right)=\frac{Ms-Md}{Md}\times 100\%$$
where S is the percentage of water absorption (%), *Ms* and *Md* are the weight (grams) of the self-healing hydrogel, which swells at time (t, *Ms*), and the initial dry hydrogel (*Md*) [23].

#### 2.4.9. Stability Study

The stability of GSNO in the PVA-B-CmChi/GSNO self-healing hydrogel was assessed under three distinct temperature conditions: room temperature (25 °C) with a relative humidity of 60%, refrigerator temperature (5 °C), and freezer temperature (−20 °C). The samples were maintained in glass vials that were fully shielded with aluminum foil to protect the self-repairing hydrogels from exposure to light. The self-healing hydrogel was subsequently broken down and placed in distilled water at predetermined time intervals using a magnetic stirrer. The clear supernatant was examined using a UV–Vis spectrophotometer (Dynamica^®^ XB-10, UK) at a wavelength of 335 nm to quantify the GSNO concentration in the PVA-B-CmChi/GSNO self-healing hydrogel [3].

### 2.5. In Vitro Drug Release

The release of NO from the PVA-B-CmChi/GSNO was examined using a dialysis membrane. The PVA-B-CmChi/GSNO self-healing hydrogel was inserted into a cellophane dialysis membrane. The release medium comprised 100 mL of PBS (pH 7.4) in the Duran bottle. Next, the membrane was placed in the release medium and the temperature was set at 37 °C, accompanied by orbital shaking at a speed of 100 rpm. At a predetermined time, one milliliter of the solution was sampled and immediately replaced with 1 mL of new PBS (pH 7.4). Then, the drug loadings were measured using a UV–Vis spectrophotometer (Dynamica^®^ Halo XB-10, Livingston, United Kingdom) at a wavelength of 335 nm. The percentage of GSNO released from the self-healing hydrogels at each time point ([NO]t) was calculated using the following equation, and the drug release kinetics were examined using DDsolver software [24]:$$\mathrm{Qn}=\frac{Cn\times Vo+\sum_{i-1}^{n-1}Ci\times Vi}{A}\times 100\%$$
where Q is % release at time n; *Cn* is drug concentration at the *n*th sampling time; *Vo* is volume of release medium; $\sum_{i-1}^{n-1}Ci\times Vi$ is total drug concentration in sampling before time *n* (correction factor); *Vi* is sampling volume; and A = amount of drug in the tested self-healing hydrogel.

### 2.6. Statistical Analysis

GraphPad Prism 8.0 was used for statistical analysis, which included one-way ANOVA and Bonferroni multiple comparison tests, as well as an unpaired *t*-test. In the event of significant *t*-test deviations, the nonparametric Mann–Whitney U tests were used to compare the distributions of two unpaired groups. *p* < 0.05 was considered statistically significant. The data are presented as means ± SD.

## 3. Results

### 3.1. Preparation of the PVA-B-CmChi and PVA-B-CmChi/GSNO Self-Healing Hydrogels

The self-healing hydrogels PVA-B-CmChi and PVA-B-CmChi/GSNO were prepared by crosslinking polymer (PVA-CmChi) and borax. The CmChi was added to improve the mechanical properties of the self-healing hydrogel. In this study, we used the Taguchi orthogonal array design to optimize the self-healing hydrogel formula. The Taguchi method is an experimental design technique used to select and regulate items or processes in order to achieve more consistency and optimization. It aims to minimize the impact of external influences and noise, resulting in less variation [25]. Three different self-healing hydrogels were made with different amounts of PVA, borax, and CmChi; namely, F1 (4%:0.8%:1.25%), F4 (4%:1.2%:1.25%), and F7 (4%:1.6%:1.25%). After polymer and GNSO were mixed, these self-healing hydrogels had a smooth surface and were evenly distributed. The physical appearance of the PVA-B-CmChi/GSNO self-healing hydrogels is shown in Figure 1A–C. The self-healing hydrogels were round-shaped and possessed a distinct pink color characteristic of GSNO. In addition, all self-healing hydrogels showed a three-dimensional network structure, which is a defining feature of hydrogels.

### 3.2. Physicochemical Characterization of the Self-Healing Hydrogel

#### 3.2.1. Morphological Analysis of Self-Healing Hydrogel PVA-B-CmChi/GSNO

The SEM images of PVA-B-CmChi/GSNO self-healing hydrogels with three different magnifications are shown in Figure 1D–L. The self-healing hydrogels were freeze-dried before SEM observation. All images showed a porous network structure with different pore sizes for each formula. Self-healing hydrogels of F1 (4%:0.8%:1.25%) and F7 (4%:1.6%:1.25%) had a denser and more irregular porous structure as compared to F4 (4%:1,2%:1.25%), which had more pores with larger sizes.

#### 3.2.2. FTIR Characterization of Self-Healing Hydrogel

The FTIR spectra of PVA, borax, and CmChi are shown in Figure 2A, whereas Figure 2B shows the FTIR spectra of PVA-B-CmChi and PVA-B-CmChi/GSNO self-healing hydrogels. The typical infrared spectrum of PVA has the following absorption bands: the broad bands at 3439.08, 1435.04, 1095.57 cm^−1^ (O–H stretching), and 2924.09 cm^−1^ (C–H stretching). The spectrum of borax showed strong bands at 945.12–1134.14 cm^−1^ (B–O). The spectrum of CmChi has the following wavenumbers; medium bands at 1604.77 cm^−1^ (N–H stretching); strong bands at 1066.64 cm^−1^ (C–OH stretching); medium bands at 1413.82 cm^−1^ (COO^−^); and medium bands at 3452.58 cm^−1^ (O–H stretching). The infrared spectrum of GSNO has the NH_2_ stretching at 3379.29 cm^−1^, strong bands at 1653.85 cm^−1^ (C=O stretching), and strong bands at 1656.85 cm^−1^ (N=O). The FTIR spectra of blank and NO-releasing self-healing hydrogels were also examined. The bands at 1433–1344 cm^−1^ and 835.18 cm^−1^ showed B–O–C and BO_3_^−^ functional groups, respectively. The NO-releasing self-healing hydrogels still also showed the band at 1512.19–1544.98 cm^−1^ that represents the N=O functional group. 

#### 3.2.3. Self-Healing Behavior of PVA-B-CmChi and PVA-B-CmChi/GSNO

Self-healing ability is the main characteristic of self-healing hydrogels, which is closely related to their application, especially in resisting external damage. Figure 3A shows the self-healing behavior of PVA-B-CmChi (F1B, F4B, and F7B) and PVA-B-CmChi/GSNO (F1, F4, and F7). The hydrogel divided into two parts was reunited/healed and can also be stretched without any cracks on the surface of the self-healing hydrogel. The time taken to be self-healed is shown in Figure 3B, which shows that F7 took a longer time to heal as compared to F1 and F4, with a healing time of 40.27 min. The healing times for F1 and F4 were 13.95 min and 26.40 min, respectively. The healing times for blank self-healing hydrogels were similar, except F7B had a longer healing time at 48.22 min as compared to its counterpart F7.

#### 3.2.4. Mechanical Characterization of Self-Healing Hydrogel

The mechanical property measurements performed resulted in the values of tensile stress, tensile strain, and Young’s modulus. The stress–strain behavior of PVA-B-CmChi and PVA-B-CmChi/GSNO self-healing hydrogels is shown in Figure 4, and the Young’s modulus values can be seen in Table 2. The tensile stress of blank self-healing hydrogels (F1B, F4B, and F7B) was 6.50, 9.50, and 15.50 swf, respectively. Similarly, the tensile strength of NO-releasing self-healing hydrogels (F1, F4, and F7) was 1.25, 5.00, and 12.75 kPa. The tensile strain for F1B, F4B, and F7B was 712.5%, 825.0%, and 975.0%, respectively, and for F1, F4, and F7 was 530.0%, 695.0%, and 837.5 (%). In addition, the Young’s modulus values were 0.91, 1.15, and 1.59 kPa for F1B, F4B, and F7B, respectively. The Young’s modulus values for F1, F4, and F7 were 0.24, 0.72, and 1.52 kPa.

#### 3.2.5. pH Characterization of Self-Healing Hydrogel

The pH of blank self-healing hydrogels (F1B, F4B, and F7B) and NO-releasing self-healing hydrogels (F1, F4, and F7) is shown in Table 2, with pH values close to neutral. There were no significant differences between the pH values of blank self-healing hydrogels and NO-releasing PVA-B-CmChi/GSNO self-healing hydrogels (*p* > 0.05). The pH of PVA-B-CmChi with three different formulations was 7.81, 7.87, and 7.91 for F1B, F4B, and F7B, respectively. Similarly, the pH of F1, F4, and F7 was 7.39, 7.54, and 7.69.

#### 3.2.6. Loading Efficiency and Loading Capacity

Table 2 shows the loading efficiency and loading capacity of NO-releasing self-healing hydrogels (PVA-B-CmChi/GSNO) from three different formulations, namely, F1, F4, and F7. The loading efficency of F1, F4, and F7 was 90.61%, 93.81%, and 97.34%. The loading capacity was 36.24%, 37.53%, and 38.94% for F1, F4, and F7, respectively.

#### 3.2.7. Swelling Ratio of Self-Healing Hydrogel

The self-healing hydrogels expanded due to the absorption of the SWF, which enhanced the release of GSNO. The PVA-B-CmChi and PVA-B-CmChi/GSNO self-healing hydrogels are quickly taken in by the SWF, reaching roughly more than 30% of its own mass in the SWF within 5 h (Figure 5). Following the swift initial absorption of SWF, the uptake of fluid by self-healing hydrogels reached maximum fluid and turned the self-healing hydrogels watery, with no notable further fluid absorption observed thereafter. The fluid absorption patterns of PVA-B-CmChi exhibited no notable distinctions compared to PVA-B-CmChi/GSNO (*p* > 0.05).

#### 3.2.8. Stability Study

The stability of GSNO in self-healing hydrogel was tested under three different temperature conditions, namely refrigerator temperature (−20 °C), 5 °C, and room temperature (25 °C with 60% relative humidity) for a 28-day period. There was a significant change in the GSNO loading in the self-healing hydrogels at three different temperatures, as shown in Figure 6. The decomposition rate of GSNO was faster at 25 °C, and it was completely decomposed at day 28. Within 28 days, about half of the GSNO loading in the self-healing hydrogels was lost at −20 °C and more than 70% at 5 °C.

### 3.3. In Vitro Drug Release

The NO-releasing self-healing hydrogels demonstrated a sustained release of NO without any sudden initial surge, as seen in Figure 7. Approximately 25% of NO was released within the initial 12-h period, and 30% was released within 24 h. There was not much difference in the release profiles for F1, F4, and F7. The kinetics of drug release followed the Korsmeyer–Peppas model, with an average R^2^ = 0.8363 (Table 3).

## 4. Discussion

As an innovative wound-dressing product, an NO-releasing self-healing hydrogel composed of PVA and CmChi as polymers, borax as a reversible crosslinking agent, and GSNO as an NO donor was developed, characterized, and evaluated. PVA is a water-soluble polymer renowned for its exceptional biocompatibility. It has found extensive application as the primary constituent in hydrogels designed to withstand tension and strain [26]. Borax is employed as a crosslinking agent in order to facilitate the formation of a hydrogel network through the binding of PVA molecules [27]. However, the use of PVA–borax results in self-healing hydrogels with inadequate mechanical properties, which hinders their application, particularly as wound dressings. Therefore, the addition of CmChi as a second polymer is expected to overcome the limitations of PVA–borax self-healing hydrogels, as CmChi can enhance the mechanical properties of self-healing hydrogels.

The use of the orthogonal Taguchi design in Table 1 led to the optimization of the formula, resulting in nine PVA-B-CmChi/GSNO formulae with different amounts of borax, CmChi, and GSNO. The ultimate objective of factorial design is to maximize the impact of formulation variables through a limited number of tests and also to determine the possibility of utilizing various types and concentration ratios of polymers and GSNO [28]. The self-healing hydrogels were successfully developed with different amounts of PVA, borax, and CmChi; namely, F1 (4%:0.8%:1.25%), F4 (4%:1.2%:1.25%), and F7 (4%:1.6%:1.25%). These hydrogels exhibited a consistency similar to that of hydrogels and had a pink color. The freeze-drying process induces phase separation and sublimation by removing water, resulting in the formation of a porous three-dimensional structure in the self-healing hydrogel [29]. This morphology was detected using scanning electron microscopy (SEM) and is depicted in Figure 1. When applied, the porous tissue structure greatly enhances the wound-healing process by facilitating fluid absorption in the wound, thereby maintaining a moist environment [30]. The morphological structure of the PVA-B-CmChi/GSNO self-healing hydrogel is influenced by varying concentrations of borax used as a crosslinking agent. Formula (F4%:1.2%:1.25%) yields a morphology characterized by a bigger and more uniform network structure compared to Formula F7 (4%:1.6%:1.25%), which has a higher concentration of crosslinkers, leading to a significantly denser and more irregular network structure. Increasing the concentration of borax causes the molecular chain to become more compact, resulting in a decrease in the distance between polymers and a denser network structure [31,32].

In order to identify the kind of crosslinking group that produces the self-healing hydrogel, FTIR analysis was used to ascertain whether functional groups were present in the hydrogel. The B–O–C crosslink, or borate ester, that results from the formation of tetrahedral complexes between PVA and borate ions is visible at the wavenumber of 1433–1344 cm^−1^. The wavenumber of 854–661 cm^−1^ represents the BO_3_-strain of the residual B(OH)_4_ from borax; however, when borax is mixed with pure PVA, certain modifications take place, such as the band widening, intensifying, and moving to higher wavenumbers [33,34]. The C=O, C–H stretching, and N–H groups are shown by the peaks at 1739.79, 1453, and 1436.97 cm^−1^, respectively. These peaks show the hydrogen bonding interaction between PVA’s –OH group and the C=O or –C–N– of CmChi. Furthermore, –CH_2_ from PVA is present at wavenumber 2924.09 cm^−1^, which changes to wavenumber 2926.01 cm^−1^, producing a stronger and larger peak in F4 following the addition of CmChi. This suggests that PVA and CmChi are crosslinked [35]. Furthermore, the FTIR spectra of the NO-releasing self-healing hydrogel (Figure 2B) show minor intensity peaks at wavelengths 1543.05 (F1), 1512.19 (F4), and 1544.98 (F7), which correspond to the N=O group of GSNO. The subtle change in band is attributed to the chemical interactions occurring between GSNO or polymers (PVA and CmChi) following the physical mixture. Emam et al. (2020) investigated the alterations in the FTIR spectra, which can manifest as either a displacement of the distinctive peak or the emergence of a novel peak [33]. When GNSO was incorporated into other materials, such as polymers, the subsequent self-healing process exhibited a notable alteration in the N=O bandwidth of the resulting GSNO. The band widens and weakens while shifting towards a longer wavelength in comparison to pure GSNO.

Wound dressings can be easily damaged by skin movement, which affects their applicability in wound healing and can cause potential secondary infection. The primary and distinctive feature of self-healing hydrogels is their capacity to repair damaged areas caused by external forces, hence achieving or even expediting the therapeutic effect [30,36]. This test was performed by halving the self-healing hydrogel, and one part was stained with methylene blue to distinguish the two pieces. The preparation was then placed at room temperature without any external stimuli, and the time taken for the two pieces to reunite was observed. The difference in self-healing time shows that the higher the concentration of borax as a crosslinking agent, the longer the time required for the preparation to reunite. This is because increasing borax tends to produce more crosslinks, which leads to a denser and more compact tissue structure, thus requiring more time to reunite [33,37]. The self-healing properties of hydrogels are a result of the combined action of physical and chemical interactions. For instance, the presence of crosslinks between borax, PVA, and CmChi enables self-healing to occur without the need for external triggers, even at room temperature [2,38].

The mechanical properties of a self-healing hydrogel influence its use, particularly in wound treatment. The higher the tensile stress–strain value, the more integrity and flexibility are assured in the self-healing hydrogel, making it easy to apply to the skin [39,40]. The tensile stress–strain value increases when the concentration of borax as a crosslinking agent increases, as weak van der Waals links are replaced by stronger covalent ones. Furthermore, increased crosslinking reduces the distance between polymer chains, making molecular chain mobility more difficult [27,37]. Young’s modulus assesses the connection between stress (applied force) and strain (resulting deformation), indicating the self-healing hydrogel’s stiffness and toughness. The Young’s modulus value is directly related to the tensile stress–strain value, which increases as the amount of borax increases. Thus, the self-healing hydrogel with the best mechanical properties is F7, which is flexible and has the highest Young’s modulus value, indicating that huge stresses do not produce deformation in the self-healing hydrogel [31,39].

A self-healing hydrogel swelling test was performed to assess its capacity to absorb wound exudate and sustain moisture in the wound region, hence accelerating the process of wound healing. Furthermore, the use of wound dressing with enhanced water retention properties can decrease the need for frequent wound dressing replacement and alleviate pain during the process [41]. Formula F7 exhibits the greatest swelling content value when compared to Formula F1 and Formula F4. The elevated concentration of borax in F7 induces greater crosslinking, resulting in reduced pore size and increased matrix density, hence enhancing water retention capabilities [33,34]. Nevertheless, the increase in water uptake of the blank self-healing hydrogels (F1B, F4B, and F7B) in the absence of GSNO was greater than that of F7. The high number of crosslinks in blank self-healing hydrogels is responsible for their denser structure, as they lack GSNO. Consequently, all FB formulations exhibit a better capacity to hold water.

Chronic wounds typically have a pH level ranging from 5.45 to 8.65. As the wound healing process progresses, the pH value gradually decreases from alkaline to neutral and eventually becomes acidic. Therefore, pH is a critical parameter that significantly affects the likelihood of successful wound healing [42,43,44]. The pH measurements of the self-healing hydrogel (Table 2) indicate that both the PVA-B-CmChi and PVA-B-CmChi/GSNO preparations have a neutral pH value. Wound dressings with a pH value of 7.4 or close to neutral are preferable since they exhibit a higher healing rate compared to wound dressings with a high alkaline pH [44]. A reduction in pH during the healing stage is essential for the growth of fibroblasts, the synthesis of DNA in cells, the supply of oxygen, the construction of collagen, the development of new blood vessels, and the activity of macrophages, all of which are important for the process of wound healing [45].

The GSNO concentration in the PVA-B-CmChi/GSNO self-healing hydrogel was then measured after the hydrogel had been stored at three distinct temperatures in order to determine the stability of GSNO in the self-healing hydrogels. GSNO is one of the NO donors, and it has a limitation; it undergoes decomposition by hydrolysis when in the presence of water. Consequently, we assessed the stability of GSNO in the self-healing hydrogels at various storage temperatures (−20, 5, and 25 °C). The loading capacity and loading efficacy of the self-healing hydrogel were assessed prior to its storage (Table 2). Borax, when utilized as a crosslinking agent, contributed to an increase in both loading capacity and loading efficiency. The degree of crosslinking increases with increasing borax concentration, thereby facilitating the entrapment of a greater quantity of drugs during the drug-loading procedure [46]. GSNO decomposed by approximately 50% at −20 °C on the 28th day of storage compared to its initial state. GSNO concentrations remained at 30% in F7, 25% in F4, and 20% in F1 at 5 °C. All the F1, F4, and F7 contained completely decomposed GSNO at room temperature. However, the GSNO content stored at low temperature for a duration of 14 days can still retain more than 70% GSNO. At room temperature (25 °C), the decomposition rate of GSNO is greater than at low temperatures. The cage effect, which refers to the stabilization of GSNO within the polymer, and concentration are both factors that influence the decomposition of GSNO [22].

The release of GSNO from the self-healing hydrogel was evaluated using a dialysis membrane and PBS as the medium for release. The release of GSNO from the self-repairing hydrogel is affected by the pore size of the formulation and the concentration of the crosslinking agent. Greater pore size results in increased release of GSNO. Nevertheless, the addition of more borax as a crosslinking agent leads to a reduction in the distance between molecular chains, resulting in decreased mobility and a denser pore size [47,48]. The release kinetics of GSNO from the self-healing hydrogel can be described by the Korsmeyer–Peppas model. This model explains the release of drugs from polymer systems through diffusion, taking into account parameters such as polymer swelling, erosion, and matrix porosity [49]. The release mechanism is characterized by Fickian diffusion, with a value of *n* = 0.21 (<0.45). This diffusion process involves the swelling of the matrix, which allows the media solution to diffuse into the preparation and be absorbed by the matrix. As a result of this swelling, the active substance diffuses out of the matrix due to the concentration gradient [50].

## 5. Conclusions

In this study, we successfully developed a self-healing hydrogel PVA-B-CmChi-based wound dressing that releases GSNO. The self-healing hydrogel is formed due to crosslinking between PVA and CmChi, which provides a sustained and controlled release of GSNO following the Korsmeyer–Peppas release kinetics model with a Fickian diffusion mechanism. Increasing the concentration of borax as a crosslinking agent affects the physicochemical characteristics of self-healing hydrogels, particularly mechanical properties. The storage stability of self-healing hydrogel showed a significant decrease in GSNO content, especially during storage at room temperature; however, low temperatures can still retain more than 70% of GSNO in the self-healing hydrogels.

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
