# Peer review of "The Formulation and Characterization of Wound Dressing Releasing S-Nitrosoglutathione from Polyvinyl Alcohol/Borax Reinforced Carboxymethyl Chitosan Self-Healing Hydrogel"

_pharmaceutics, 2024, doi:10.3390/pharmaceutics16030344_

Round 1

Reviewer 1 Report

Comments and Suggestions for Authors

The manuscript entitled ‘The Formulation and Characterization of Wound Dressing Releasing S-Nitrosoglutathione from Polyvinyl Alcohol/Borax Reinforced Carboxymethyl Chitosan Self-Healing Hydrogel” presents evaluation of structural, mechanical, and physicochemical properties of polyvinyl alcohol (PVA)-based hydrogels with borax and carboxymethyl chitosan. This is a very interesting article and the technical quality of this manuscript within its field is good. Nevertheless, I have a few remarks:

1.      You wrote that “Self-healing hydrogel possesses the additional benefit of being biocompatible, biodegradable, and capable of conforming to the shape, size, and imperfections of the area where it is applied [10]”. It should be corrected since it is an incorrect statement since the term ‘self-healing’ does not mean biocompatibility and biodegradability but the self-healing property of hydrogel comes from the reversible physical or chemical bonds.

2.      Borax was used as a crosslinking agent. Did you perform cytotoxicity assays? Since borax can be toxic to human cells.

3.      There is no explanation for the abbreviations Ms and Md in the water absorption capacity equation.

4.      Could you explain exactly how you calculated the in vitro drug release? Have you taken into account the dilution of the release medium?

5.      3.1. Preparation of the PVA-B-CmChi and PVA-B-CmChi/GSNO self-healing hydrogels: In my opinion, the first statement should be verified because it may be misunderstood, namely that the PVA-borax was cross-linked by CmChi.

6.      Figure 1 is illegible because it is not clear which photo shows which samples. You wrote in the caption: “Scanning electron microscope (SEM) images of F1, F4, and F7 at 500x (D,E,F); at 1000x (G,H,I); and at 3000x.” – Thus, images J, K, L shows images at 3000x magnification?

7.      Why do you think the F4 samples had a more porous structure than the other samples?

8.      Figure 3: What type of test did you perform during statistical analysis? What does mean “*”? How much force did you apply during the stretch?

9.      3.2.4. Mechanical characterization of self-healing hydrogel: The abbreviation of kilopascals is kPa, not Kpa.

10.   3.2.7. Swelling ratio of self-healing hydrogel: Why did you stop assessing the swelling capacity of the hydrogels after 5 hours if you did not observe a plateau phase? You wrote in the Methods section that the assessment of swelling ability was performed by using distilled water. However, in the Results section, you wrote that this experiment was performed in simulated wound fluid. Please clarify this. Figure 5 is illegible.

11.   According to available literature, it was proved that dressings with an acidic nature are better than dressings with neutral pH due to their pro-healing properties, especially for chronic wound applications. In your study, hydrogels had a neutral pH. Where do you think they can find applications? Did you perform biodegradation in an acidic pH? The wounds may have e.g. 5.5 pH. What about the stability of hydrogels under such conditions?

12.   The big disadvantage of this article is the lack of cytotoxicity assessment of fabricated hydrogels since this is a crucial issue during designing biomaterials for potential biomedical applications.

Author Response

Response to reviewer’s comments

We would like to thank the reviewers for their constructive and profound comments on this manuscript. The manuscript has been revised to incorporate all the insightful comments from the reviewers. Our point-by-point responses to the reviewer’s comments are provided below. All revisions in the revised manuscript are highlighted by red color.

Reviewer #1

The manuscript entitled ‘The Formulation and Characterization of Wound Dressing Releasing S-Nitrosoglutathione from Polyvinyl Alcohol/Borax Reinforced Carboxymethyl Chitosan Self-Healing Hydrogel” presents evaluation of structural, mechanical, and physicochemical properties of polyvinyl alcohol (PVA)-based hydrogels with borax and carboxymethyl chitosan. This is a very interesting article and the technical quality of this manuscript within its field is good. Nevertheless, I have a few remarks:

*We thank the reviewer for highlighting our major findings and for the constructive comments on this manuscript. Below please find the responses to the comments. Corresponding changes have been made in the revised manuscript.

  1. You wrote that “Self-healing hydrogel possesses the additional benefit of being biocompatible, biodegradable, and capable of conforming to the shape, size, and imperfections of the area where it is applied [10]”. It should be corrected since it is an incorrect statement since the term ‘self-healing’ does not mean biocompatibility and biodegradability but the self-healing property of hydrogel comes from the reversible physical or chemical bonds.

(Response) As suggested. We have revised the sentence for clarification (Page 2, Lines 62-64).

  1. Borax was used as a crosslinking agent. Did you perform cytotoxicity assays? Since borax can be toxic to human cells.

(Response) We thank the reviewer’s insightful comment. We did not include the toxicity study in this manuscript. However, the toxicity test toward L929 fibroblast cell showed no toxicity either for blank self-healing hydrogel (PVA-B-CmChi) and NO-releasing self-healing hydrogels (PVA-B-CmChi/GSNO). Since borax is a crosslink agent, it forms a borate ester bond in self-healing hydrogel and thus is not free from.

In addition, Borax (Sodium Tetraborate) is not acutely toxic, based on several study. The chemical demonstrates that a substantial amount of the substance is required to induce severe symptoms or mortality, as evidenced by its LD 50 (median lethal dose) value of 2.66 g/kg in rats. Another study also mentioned the majority of toxicity data on boron are on boric acid. Borax and boric acid are distinct formulations of the same compound-boron. Sodium Tetraborate (Na₂B₄O₇*10H₂O) constitutes borax, which is composed of boron, oxygen, and sodium. Boric acid is generated through a combination of borax and various naturally occurring minerals, including colemanite and boracite.

Hadrup, N., Frederiksen, M., & Sharma, A. K. (2021). Toxicity of boric acid, borax and other boron containing compounds: A review. Regulatory toxicology and pharmacology, 121, 104873.

  1. There is no explanation for the abbreviations Ms and Md in the water absorption capacity equation.

(Response) As suggested, we have added the information related to the Ms and Md in water absortion capacity (Page 6, Line 198).

  1. Could you explain exactly how you calculated the in vitro drug release? Have you taken into account the dilution of the release medium?

(Response) We calculated the in vitro drug release according to the general method, following the below equation. Because we used the sink condition, a correction factor is needed.

Q = % release at time n

Cn= drug concentration at the nth sampling time

Vo= volume of release medium

 = Total drug concentration in sampling before time n (correction factor)

S = sampling volume

A = amount of drug in the tested self-healing hydrogel

  1. 1. Preparation of the PVA-B-CmChi and PVA-B-CmChi/GSNO self-healing hydrogels: In my opinion, the first statement should be verified because it may be misunderstood, namely that the PVA-borax was cross-linked by CmChi.

(Response) As suggested, we have revised the sentence (Page 6, Line 231)

  1. Figure 1 is illegible because it is not clear which photo shows which samples. You wrote in the caption: “Scanning electron microscope (SEM) images of F1, F4, and F7 at 500x (D,E,F); at 1000x (G,H,I); and at 3000x.” – Thus, images J, K, L shows images at 3000x magnification?

(Response) We thank the reviewer’s correction. We have added information regarding images J, K, L in the caption of Figure 1 (Page 7, Line 251).

  1. Why do you think the F4 samples had a more porous structure than the other samples?

(Response) We thank the reviewer’s comment. In our study, all self-healing hydrogel (F1, F4, and F7) produces porous structure. However, F1 and F7 had a denser and more irregular porous structure as compared to F4 which had more porous with larger size. This might be due to the increasing concentration of borax in the PVA/borax ratio will increase the occurrence of cross-links formed, and the pores formed will also be more even, resulting in a denser and more compact structure. From the result (Figure 1) we can see that the concentration of borax (F4, 1.2%) is probably the optimal concentration to produce preferable pores structure with larger size. Furthermore, it is recognized that chemical cross-linking agents have a drawback in that they can induce the creation of non-uniform cross-links as a result of uneven distribution, leading to the accumulation of cross-links in specific regions of the hydrogel, thereby causing the presence of large pores in other regions.

  1. Figure 3: What type of test did you perform during statistical analysis? What does mean “*”? How much force did you apply during the stretch?

(Response) We used one-way anova statistical analysis to test differences between groups. The "*" in the Figure 3B, indicates that F7B vs F7 is significantly different (P< 0.05).

The mechanical properties test using an electronic universal testing machine at a crossheaving speed of 100 mm min-1 and the load force varied from 0.5-6.2 Newton.

  1. 2.4. Mechanical characterization of self-healing hydrogel: The abbreviation of kilopascals is kPa, not Kpa.

(Response) We thank for pointing out the mistake. We have corrected the typos (page 9, Lines 294-297).

  1. 2.7. Swelling ratio of self-healing hydrogel: Why did you stop assessing the swelling capacity of the hydrogels after 5 hours if you did not observe a plateau phase? You wrote in the Methods section that the assessment of swelling ability was performed by using distilled water. However, in the Results section, you wrote that this experiment was performed in simulated wound fluid. Please clarify this. Figure 5 is illegible

(Response) We appreciate the reviewer’s correction. We have revised the method part and figure for clarification. We drew the figure for up to 5 hours because, after that, some hydrogels uptake more liquid (SWF) and lose their 3D hydrogel structure and therefore cannot maintain the initial dry hydrogel weight (Md). Which then could not be included in the equation.

  1. According to available literature, it was proved that dressings with an acidic nature are better than dressings with neutral pH due to their pro-healing properties, especially for chronic wound applications. In your study, hydrogels had a neutral pH. Where do you think they can find applications? Did you perform biodegradation in an acidic pH? The wounds may have e.g. 5.5 pH. What about the stability of hydrogels under such conditions?

(Response) Chronic wounds are in the pH range of 5.45 to 8.65. We believe that our hydrogel at pH around 7 can be used as a wound dressing that does not worsen or inhibit the wound healing process. In addition, our active ingredient, S-nitrosoglutathione (GSNO), is a wound healing enhancer. Many studies have proved that nitric oxide (NO) accelerates wound healing. Thus, our self-healing hydrogel is the platform for NO released from GSNO. Moreover, once GSNO is released from the self-healing hydrogel, it will decompose into NO and Glutathione, having a pH of acidic in the aqueous solution.

We did not perform a biodegradation study at an acidic pH or check the stability of our hydrogels under those conditions in the current study. However, we are planning to do it in the future.

  1. The big disadvantage of this article is the lack of cytotoxicity assessment of fabricated hydrogels since this is a crucial issue during designing biomaterials for potential biomedical applications.

(Response) We thank the reviewer’s comment. The present investigation is centered around the development and analysis of self-healing hydrogels that release nitric oxide (NO). Therefore, we omitted the evaluation of cytotoxicity. Nevertheless, we want to proceed with in vivo wound healing and then incorporate the toxicity investigation.

Reviewer 2 Report

Comments and Suggestions for Authors

 In the manuscript entitled “The Formulation and Characterization of Wound Dressing Releasing S-Nitrosoglutathione from Polyvinyl Alcohol/Borax Reinforced Carboxymethyl Chitosan Self-Healing Hydrogel” Hasan and co-workers describe a new self-healing wound dressing material made of reversibly borax-crosslinked polyvinyl alcohol hydrogel, further reinforced with carboxymethyl chitosan, and loaded with the NO generating compound S-Nitrosoglutathione.

The work is well designed and performed competently, to good technical and scientific standards. The experimental methodologies are described in enough detail to warrant reproduction by interested readers. The authors make use of a robust ensemble of techniques to characterize their wound dressing material regarding self-healing properties and to demonstrate its S-Nitrosoglutathione-releasing properties.  The results are well presented and thoroughly discussed.

I believe that this manuscript will be appealing to the broad community of researchers interested in biomaterials and drug release. The manuscript is certainly worth publication. The authors could raise significantly the profile of their work by including some biological (preliminary) data in the manuscript.  

As a minor issue, the authors must describe in better detail the “Preparation of self-healing hydrogels” in section 2.3, specifying the amounts of reagents used in the preparation of their hydrogels to give readers an idea of the scale of the synthesis.

Best regards

Author Response

Reviewer #2

In the manuscript entitled “The Formulation and Characterization of Wound Dressing Releasing S-Nitrosoglutathione from Polyvinyl Alcohol/Borax Reinforced Carboxymethyl Chitosan Self-Healing Hydrogel” Hasan and co-workers describe a new self-healing wound dressing material made of reversibly borax-crosslinked polyvinyl alcohol hydrogel, further reinforced with carboxymethyl chitosan, and loaded with the NO generating compound S-Nitrosoglutathione.

The work is well designed and performed competently, to good technical and scientific standards. The experimental methodologies are described in enough detail to warrant reproduction by interested readers. The authors make use of a robust ensemble of techniques to characterize their wound dressing material regarding self-healing properties and to demonstrate its S-Nitrosoglutathione-releasing properties.  The results are well presented and thoroughly discussed.

I believe that this manuscript will be appealing to the broad community of researchers interested in biomaterials and drug release. The manuscript is certainly worth publication. The authors could raise significantly the profile of their work by including some biological (preliminary) data in the manuscript.  

As a minor issue, the authors must describe in better detail the “Preparation of self-healing hydrogels” in section 2.3, specifying the amounts of reagents used in the preparation of their hydrogels to give readers an idea of the scale of the synthesis.

*We thank the reviewer for the positive comments and for the constructive comments on this manuscript. Corresponding changes have been made in the revised manuscript for the minor issues (Page 3, Line 127-132).

Reviewer 3 Report

Comments and Suggestions for Authors

Comment to Author

I read the paper entitled "The Formulation and Characterization of Wound Dressing Re- leasing S-Nitrosoglutathione from Polyvinyl Alcohol/Borax Re-inforced Carboxymethyl Chitosan Self-Healing Hydrogel" done by Palungan et al. The manuscript looks interesting and this is a worthwhile subject. In addition to the well-organized research paper, the author's presentation of the results and the drawn figures is impressive. However, some major and minor concerns exist regarding this paper as following. The following point should be added to the revised manuscript and changed.

Please modify this in the entire manuscript.

1.       Introduction section.

o   What are your studies aim, objectives and hypotheses of your study? They should be included in the last paragraph of the introduction.

2.       Materials method section.

o   Page 3, line 110. “Polyvinyl alcohol (PVA, 110 MW=44,05, 87-89%)”what is the molecular weight, correct it.

o   Provide purity and specifications for each material, such as carboxymethyl chitosan.

 Section 2.5.

o   I suggest that the author take a photo of how to measure the release of drug through the dialysis tube. For readers of this paper, it will be helpful to understand the method and apply it to their research.

3.       Results and discussion

o   It would be helpful if the author discussed the limitations of their experiment as well in the discussion section.

o   Have you compared the zeta potential of your materials with the zeta potential of the skin? It is recommended that the author conduct this experiment and include it in the revised manuscript. Your paper will receive an additional bonus as a result of this.

Author Response

Reviewer #3

Comment to Author

I read the paper entitled "The Formulation and Characterization of Wound Dressing Re- leasing S-Nitrosoglutathione from Polyvinyl Alcohol/Borax Re-inforced Carboxymethyl Chitosan Self-Healing Hydrogel" done by Palungan et al. The manuscript looks interesting and this is a worthwhile subject. In addition to the well-organized research paper, the author's presentation of the results and the drawn figures is impressive. However, some major and minor concerns exist regarding this paper as following. The following point should be added to the revised manuscript and changed.

Please modify this in the entire manuscript.

*We thank the reviewer for highlighting our major findings and for the constructive comments on this manuscript. Below please find the responses to the comments. Corresponding changes have been made in the revised manuscript.

  1. Introduction section.

o   What are your studies aim, objectives and hypotheses of your study? They should be included in the last paragraph of the introduction.

(Response) As suggested, we have revised the introduction in the revised manuscript (Page 3, Lines 94-99)

  1. Materials method section.

o   Page 3, line 110. “Polyvinyl alcohol (PVA, 110 MW=44,05, 87-89%)”what is the molecular weight, correct it.

 (Response) We have corrected the typos (Page 3, Line 108)

o   Provide purity and specifications for each material, such as carboxymethyl chitosan.

 Section 2.5.

(Response) We have added the requested information (page 3, Line 109)

o   I suggest that the author take a photo of how to measure the release of drug through the dialysis tube. For readers of this paper, it will be helpful to understand the method and apply it to their research.

 (Response) We thank the reviewer’s suggestion; we have revised the method part for clarification (Page 6, line 215-221).

  1. Results and discussion

o   It would be helpful if the author discussed the limitations of their experiment as well in the discussion section.

(Response) As suggested we have added the limitation in this study in the revised manuscript (Page 14, lines 444-447)

o   Have you compared the zeta potential of your materials with the zeta potential of the skin? It is recommended that the author conduct this experiment and include it in the revised manuscript. Your paper will receive an additional bonus as a result of this.

(Response) We thank the reviewer’s suggestion. However, our current study is centered around the development and analysis of self-healing hydrogels that release nitric oxide (NO). Therefore, we did not evaluate the zeta potential of our self-healing hydrogels and compare it with the skin’s. Nevertheless, we would like to add those studies in the future.

Reviewer 4 Report

Comments and Suggestions for Authors

1.      Specify the GSNO acronym in the abstract

2.      Use superscripts for power numbers (i.e. -1, line 178)

3.      How does the presence of CmChi affect the pore size? Is it possible to report the degree of porosity (%)? What is the meaning of yellow arrows?

4.      Are the mechanical performances compatible with the targeted application? How does the CmChi improve the tensile stress?

5.      Similarly, why did the LE and LC increase from F1 to F7? Could it be due to the formation of a tight network?

6.      Figure 5: the data are hard to read since the data are overlapped

7.      Based on the data reported in Figure 6, is it possible to claim that the prepared hydrogel is stable over time as T changes? It seems that there are no so noticeable differences between very low T and room T. Which should be the specific time within which the materials should be applied? Then, should the drug loafing reduction be considered critical? This section should be improved and better discussed.

8.      If the hydrogel is stable up to 28 days, why did the authors perform the release for 24 h? What about the remaining drug amount of 70%?

Comments on the Quality of English Language

Moderate editing of English language required

Author Response

Reviewer #4

  1. Specify the GSNO acronym in the abstract

(Response) As suggested, we have added the acronym GSNO in the abstract (Page 1, Line 21).

  1. Use superscripts for power numbers (i.e. -1, line 178)

(Response) As suggested, we have added superscripts for power numbers in the revised manuscript (Page 5, Line 167).

  1. How does the presence of CmChi affect the pore size? Is it possible to report the degree of porosity (%)? What is the meaning of yellow arrows?

(Response) In our study, as shown in Figure 1, the CmChi concentration in the PVA:B:CmChi ratio of F1 (4%:0.8%:1.25%), F4 (4%:1.2%:1.25%), and F7 (4%:1.6%:1.25%) were all the same 1.25%, meaning that the concentration of CmChi itself did not affect the pore size. What truly affects the pore structure and size is the degree of crosslinking; thus, borax concentration as a crosslinking agent is crucial.

The yellow arrows denote the pore of self-healing hydrogels. We have added this information in the caption of Figure 1 (page 7, Line 254).

  1. Are the mechanical performances compatible with the targeted application? How does the CmChi improve the tensile stress?

(Response) We thank the reviewer’s comment. We believe that the self-healing hydrogel produced possesses mechanical properties that are suitable for the intended use. Our self-healing hydrogel possesses pliable characteristics and is well-suited for use as a wound dressing. This is demonstrated by the examination of the mechanical characteristics using high tensile strain and Young's modulus values, ensuring that the self-healing hydrogel remains undistorted even under significant stress. The combination of PVA and borax by borate ester cross-linking results in the formation of a rigid hydrogel. Furthermore, the incorporation of CmChi leads to the creation of a dual network by cross-linking PVA and CmChi. The hydroxyl groups present in PVA and the hydroxyl, amino, and carboxyl groups present in CmChi have the ability to establish intermolecular hydrogen bonds. The incorporation of CmChi results in the formation of a self-healing hydrogel that exhibits superior mechanical characteristics compared to PVA-borax self-healing hydrogels.

  1. Similarly, why did the LE and LC increase from F1 to F7? Could it be due to the formation of a tight network?

(Response) The increase in LC and LE from F1 to F7 is likely due to increased cross-linking, resulting in a denser network structure that allows more GSNO to be trapped during the drug loading process.

  1. Figure 5: the data are hard to read since the data are overlapped

(Response) We have made a small revision to Figure 5 for readability (Page 11, Line 324).

  1. Based on the data reported in Figure 6, is it possible to claim that the prepared hydrogel is stable over time as T changes? It seems that there are no so noticeable differences between very low T and room T. Which should be the specific time within which the materials should be applied? Then, should the drug loafing reduction be considered critical? This section should be improved and better discussed.

(Response) We thank the reviewer for the insightful comment. Figure 6 is related to the GSNO stability in the self-healing hydrogels. It doesn’t necessarily mean hydrogel stability. GSNO is one of NO donors and decomposes by hydrolysis in the presence of water. Thus, we evaluated the GSNO stability in the self-healing hydrogels under different storage temperatures (-20, 5, and 25 oC). Figure 6 shows that among three different storage conditions, GSNO in the self-healing hydrogel was less decomposed at cold temperatures (-20 and 5oC) than at room temperature (25oC), which showed all GSNO was decomposed at day 28. The results also suggested that the GSNO content in the self-healing hydrogels storage at -20oC remains more than 75% at day 7, which is still sufficient for use in wound healing. Yes, the drug loading reduction is considered critical, because the drug loading affects the wound healing activity of GSNO.

  1. If the hydrogel is stable up to 28 days, why did the authors perform the release for 24 h? What about the remaining drug amount of 70%?

(Response) We evaluated the stability of GSNO in the self-healing hydrogels under different storage conditions for 28 days. This study was to evaluate the best storage condition of our NO-releasing self-healing hydrogels. This study is different from the NO release study. A NO release study was conducted under 37 oC in the release medium to examine the release profile of GSNO from self-healing hydrogels. Because most wound dressings need to be frequently changed daily, we would like to see the sustained release profile of our drug in the self-healing hydrogels. Since our release results showed a sustained release profile, we could later reduce the frequency of dressing changes. Frequency dressing changes are associated with an increased incidence of wound sepsis and delayed wound healing.

Round 2

Reviewer 1 Report

Comments and Suggestions for Authors

Thank the Authors for the revision. All my suggestions were addressed. I accept the manuscript in its present form.

Reviewer 3 Report

Comments and Suggestions for Authors

The author has addressed all comments in the revised manuscript. After revision, the manuscript appears to be in good shape. 

Reviewer 4 Report

Comments and Suggestions for Authors

The manuscript has been improved according to the proposed revisions. I can now recommend the publication.